# Balancing elementary steps enables coke-free dry reforming of methane

Jiaqi Yu [1,8], Tien Le [2], Dapeng Jing [3], Eli Stavitski [4], Nicholas Hunter[5], Kanika Lalit[1], Denis Leshchev[4], Daniel E. Resasco [2], Edward H. Sargent [6,7] ✉, Bin Wang [2] ✉ & Wenyu Huang [1] ✉

Balancing kinetics, a crucial priority in catalysis, is frequently achieved by sacrificing activity of elementary steps to suppress side reactions and enhance catalyst stability. Dry reforming of methane (DRM), a process operated at high temperature, usually involves fast C-H activation but sluggish carbon removal, resulting in coke deposition and catalyst deactivation. Studies focused solely on catalyst innovation are insufficient in addressing coke formation efficiently. Herein, we develop coke-free catalysts that balance kinetics of elementary steps for overall thermodynamics optimization. Beginning from a highly active cobalt aluminum oxide ($CoAl_2O_4$) catalyst that is susceptible to severe coke formation, we substitute aluminum (Al) with gallium (Ga), reporting a $CoAl_{0.5}Ga_{1.5}O_4$-R catalyst that performs DRM stably over 1000 hours without observable coke deposition. We find that Ga enhances DRM stability by suppressing C-H activation to balance carbon removal. A series of coke-free DRM catalysts are developed herein by partially substituting Al from $CoAl_2O_4$ with other metals.

Heterogeneous catalysis involves a close interplay between two important disciplines: chemical kinetics and thermodynamics. Improving reaction kinetics and catalytic stability are crucial priorities in modern catalytic society. Elevated temperature is not only important for achieving a high reaction rate to meet the requirement of industrial processes but also necessitated for endothermic reactions to bias the equilibrium toward products[1]. However, at high temperatures, reaction flux is already pronounced and dissipation of the thermal energy into the surface species becomes challenging to control, driving undesirable reactions that often lead to catalysts deactivation. In such cases, reaction thermodynamic control in terms of minimizing side reactions and improving catalysis stability becomes extremely important and is often pursued at the price of kinetics.

Dry reforming of methane (DRM), the reaction between two greenhouse gases, methane and carbon dioxide, to produce syngas, is such an endothermic catalytic process operated at elevated temperatures (typically above 650 °C). Balanced kinetics between two coupled elementary steps, methane dehydrogenation and carbon removal, is critical in DRM to reach decent conversion and maintain a durable catalytic process. With numerous efforts devoted to catalysts development over several decades, catalysts, especially non-precious Ni- and Co-based catalysts[2–5], have approached good activities reaching the equilibrium conversion but face severe coke deposition due to the unbalanced kinetics between $CH_4$ activation and carbon removal[6]. Depositing carbon in any format could deactivate the catalyst, break the catalytic pellet, and block the reactor. Thus, preventing coke

[1]Department of Chemistry, Iowa State University, Ames, IA 50011, USA. [2]School of Sustainable Chemical, Biological and Materials Engineering, University of Oklahoma, Norman, OK 73019, USA. [3]Materials Analysis and Research Laboratory, Iowa State University, Ames, IA 50010, USA. [4]National Synchrotron Light Source II, Brookhaven National Laboratory, Upton, NY 11973, USA. [5]Department of Mechanical Engineering, Iowa State University, Ames, IA 50011, USA. [6]Department of Chemistry, Northwestern University, Evanston, IL 60208, USA. [7]Department of Electrical and Computer Engineering, Northwestern University, Evanston, IL 60208, USA. [8]Present address: Department of Chemistry, Northwestern University, Evanston, IL 60208, USA. ✉e-mail: ted.sargent@northwestern.edu; wang_cbme@ou.edu; whuang@iastate.edu

formation and improving catalyst stability have become the crux of the matter in DRM.

Novel catalyst design approaches, such as protecting active species[7–10], optimizing metal-support interaction[11–15], developing single-atom active sites[16–18], and doping heteroatoms[19–21], have been reported to enhance catalytic stability. However, the coke-free DRM at the industrial scale remains hardly accessible. Furthermore, the developed strategies tend to center on specific catalyst structures or dispersions and often require sophisticated synthesis processes, thereby encountering obstacles when it comes to practical applications. Although system modifications have been demonstrated to indirectly enhance coke elimination, for example, by adding extra $CO_2$[22], combining with direct methane oxidation (adding $O_2$) and steam reforming (adding $H_2O$)[23,24], using light-driven DRM[25,26] and substituting $CO_2$ with $H_2S$[27], it is still a great challenge to develop a principle or guideline for the rational design of robust and coke-free DRM.

Herein, we utilize comprehension of the fundamentals in heterogeneous catalysis to introduce a concise and generalizable concept for designing coke-free catalysts that relies on the interplay between chemical kinetics and dynamics. We envision that balancing kinetics of elemental steps, activation of C-H and carbon removal, could lead to overall thermodynamics optimization (illustrated in Supplementary Fig. 1). As a proof of principle, combining a previously reported active DRM catalyst $CoAl_2O_4$ with DRM non-active $CoGa_2O_4$ yields a robust and coke-free DRM catalyst, $CoAl_{0.5}Ga_{1.5}O_4$-R. The rationally designed $CoAl_{0.5}Ga_{1.5}O_4$-R catalyst presents a stable and coke-free DRM process during a 1000 h reaction test. Experimental and theoretical studies

were conducted to examine the mechanism of this coke-free DRM. Temperature-programmed surface reactions (TPSR) rationalize the role of Ga in achieving more balanced kinetics between redox steps by reducing $CH_4$ activation to match carbon removal. Density functional theory (DFT) calculations provide additional mechanistic insights, demonstrating that Ga increases the energy required for $CH_4$ dehydrogenation and slows down the kinetic of this step while promoting carbon removal. In addition to $CoAl_{0.5}Ga_{1.5}O_4$-R, the idea of balancing kinetics for thermodynamics optimization makes it feasible to develop a series of coke-free DRM catalysts, such as $CoAlVO_4$-R, $CoAlMnO_4$-R, and $CoAlFeO_4$-R, which facilitates the catalyst design flexibility for practical applications.

## Results & discussion

### Catalyst design directed by balancing kinetics

To demonstrate the coke-free catalyst based on the compromised kinetics of dry reforming of methane (DRM) for the thermodynamics of coke formation, we started with a previously reported Co catalyst derived from spinel oxides[28]. The prepared spinel oxides were reduced at 750 °C for 2 h before the reaction, and the corresponding catalysts were named as $CoAl_xGa_{(2-x)}O_4$-R (x = 2, 1.5, 1, 0.5, 0). The DRM test was performed at 700 °C and with high space velocity ($CH_4$:$CO_2$:He=1:1:8, total gas-hourly-space-velocity (GHSV) = 300 $Lg_{cat}^{-1}h^{-1}$) to avoid the equilibrium conversion, where the catalyst stability can be truly evaluated. As seen in Fig. 1, although $CoAl_2O_4$-R is highly active for DRM and has moderate deactivation (3% conversion loss) during a 100 h stability test, significant coke deposition was detected. The Raman

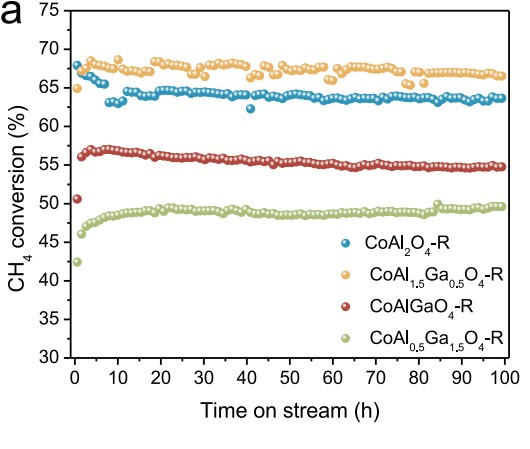

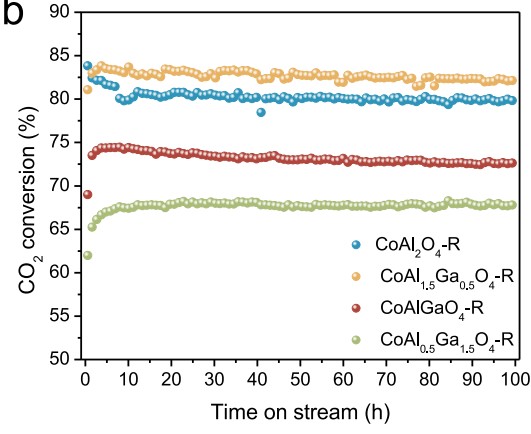

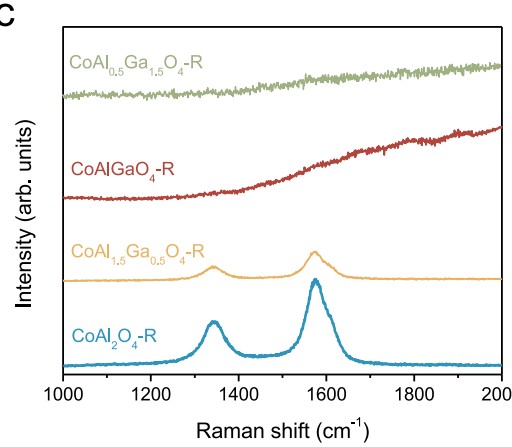

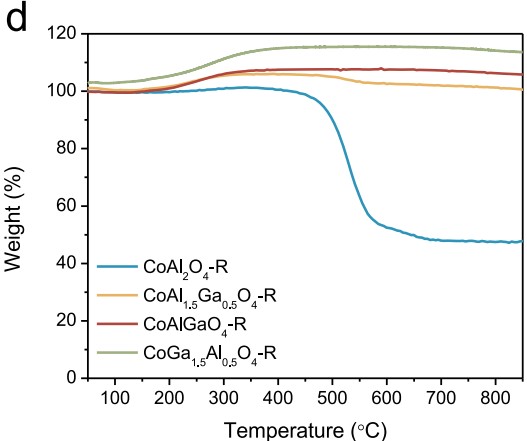

**Fig. 1 | DRM test of $CoAl_xGa_{(2-x)}O_4$-R catalysts (x = 0.5, 1, 1.5, 2).** Catalytic performance of $CoAl_xGa_{(2-x)}O_4$-R catalysts in 100 h test: **a** $CH_4$ conversion and **b** $CO_2$ conversion; 300 $Lg_{cat}^{-1}h^{-1}$ reaction gas feeding. Post-reaction (700 °C, 100 h) characterization of $CoAl_xGa_{(2-x)}O_4$-R catalysts: **c** Raman spectra and **d** TGA. Equilibrium conversion of $CH_4$ and $CO_2$ at 700 °C are 84.0% and 89.6%, respectively[43,44].

spectroscope (Fig. 1c) shows that the spent $CoAl_2O_4$-R has two intense peaks at 1345 and 1575 $cm^{-1}$, corresponding to the characteristic D and G bands of carbon. Strong $CO_2$ peaks and around 60% weight loss were observed during oxygen temperature-programmed oxidation ($O_2$-TPO, Supplementary Fig. 4a) and thermogravimetric analysis (TGA, Fig. 1d). In addition, the scanning electron microscopy (SEM) image of spent $CoAl_2O_4$-R shows catalyst particles wrapped by dense carbon nanotubes (Supplementary Fig. 4b). On the contrary, $CoGa_2O_4$-R shows no activity for DRM under the same condition.

The versatile elemental composition of the spinel oxide family makes it feasible to optimize catalysts by replacing one element with another without changing the crystal structure. We partially replaced $Al^{3+}$ with $Ga^{3+}$ to prepare $CoAl_xGa_{(2-x)}O_4$ spinel oxides and intentionally lower the activity of $CoAl_2O_4$. Fig. 1a, b show that with the replacement of Ga, $CoAl_xGa_{(2-x)}O_4$ derived catalysts ($CoAl_xGa_{(2-x)}O_4$-R) show negligible deactivation during a 100 h DRM stability test. Among them, $CoAl_{0.5}Ga_{1.5}O_4$-R showed no deactivation with a stable $CH_4$ conversion of 49% and $CO_2$ of 68%, both of which are lower than those on the $CoAl_2O_4$-R. The post-reaction Raman, $O_2$-TPO, TGA, and SEM were conducted on spent catalysts to evaluate the coke resistance properties of $CoAl_xGa_{(2-x)}O_4$-R catalysts. From Raman spectra on spent $CoAl_{1.5}Ga_{0.5}O_4$-R in Fig. 1c, the D- and G-band peaks of carbon weakened compared to $CoAl_2O_4$-R. Further decreasing Al:Ga ratio, no carbon peaks could be detected on used $CoAlGaO_4$-R and $CoAl_{0.5}Ga_{1.5}O_4$-R catalysts. The same trend was observed in $O_2$-TPO and TGA, that is, decreasing Al:Ga ratio to 1:1 and 1:3 leads to negligible $CO_2$ peak from $O_2$-TPO and no weight loss from TGA (Fig. 1d and Supplementary Fig. 4). We can thus conclude that while replacing Al with Ga makes the catalysts less active, the coke resistance property is dramatically improved.

## Coke-free catalyst evaluation

We further studied the $CoAl_{0.5}Ga_{1.5}O_4$-R catalyst evaluating its catalytic performance during over 1000 h DRM test. High resolution-scanning transmission electron microscopy (HR-STEM) was conducted to characterize the structure of catalyst before and after reduction. As shown in Fig. 2a and b, before reduction, $CoAl_{0.5}Ga_{1.5}O_4$ has a characteristic spinel oxide structure, while after reduction, the transformation of the spinel oxides to CoGa intermetallic compounds was observed from high-angle annular dark-field scanning transmission electron microscopy (HAADF-STEM), whose elemental distribution are confirmed with X-ray energy dispersive spectroscopy (EDS) mapping shown in Supplementary Fig. 5. The intermixing of the two metal atoms in CoGa intermetallic structure in $CoAl_{0.5}Ga_{1.5}O_4$-R was also confirmed by X-ray photoelectron spectroscopy (XPS) and X-ray absorption spectroscopy (XAS), as shown in Supplementary Figs. 7 and 8. A temperature-dependent catalytic evaluation was performed under a constant stream of $CH_4:CO_2:He = 1:1:8$ and total gas-hourly-space-velocity (GHSV) = 300 $Lg_{cat}^{-1}h^{-1}$. The DRM was stable at 800 °C followed by lower temperature tests at 750 and 700 °C. Lowering DRM temperature, the reaction rate decreases accordingly, while raising the temperature back to 800 °C, the reaction rate recovers completely (Fig. 2c). To explore the stability of the $CoAl_{0.5}Ga_{1.5}O_4$-R in DRM, a 1000 h stability test was performed at 700 °C with intentionally reduced conversions ($CH_4$ conversion ~40% and $CO_2$ ~ 55%). No deactivation was observed through $CH_4$ conversion, while a slight decrease in $CO_2$ conversion could be attributed to reduced reverse water-gas shift reaction (Fig. 2d)[29]. The post-reaction characterization confirms no coke deposition after the 1000 h DRM test. As shown in Fig. 2e, Raman spectrum shows no peak corresponding to any form of carbon. No weight loss of spent catalyst was observed from TGA under air, and the mass spectrometer (MS) connected to TGA did not detect $CO_2$ from coke combustion (Fig. 2f). The weight increases from 200-400 °C during TGA is caused by the oxidation of metal in the catalyst. SEM was conducted as evidence of clean catalysts' surface shown in Supplementary Fig. 9. The over 1000 h stability test and the post characterization of the used catalyst conclude that the $CoAl_{0.5}Ga_{1.5}O_4$-R is a robust and coke-free DRM catalyst with potential for industrial-scale application (stability comparison of $CoAl_{0.5}Ga_{1.5}O_4$-R and state-of-the-art DRM catalysts shown in Supplementary Table 2).

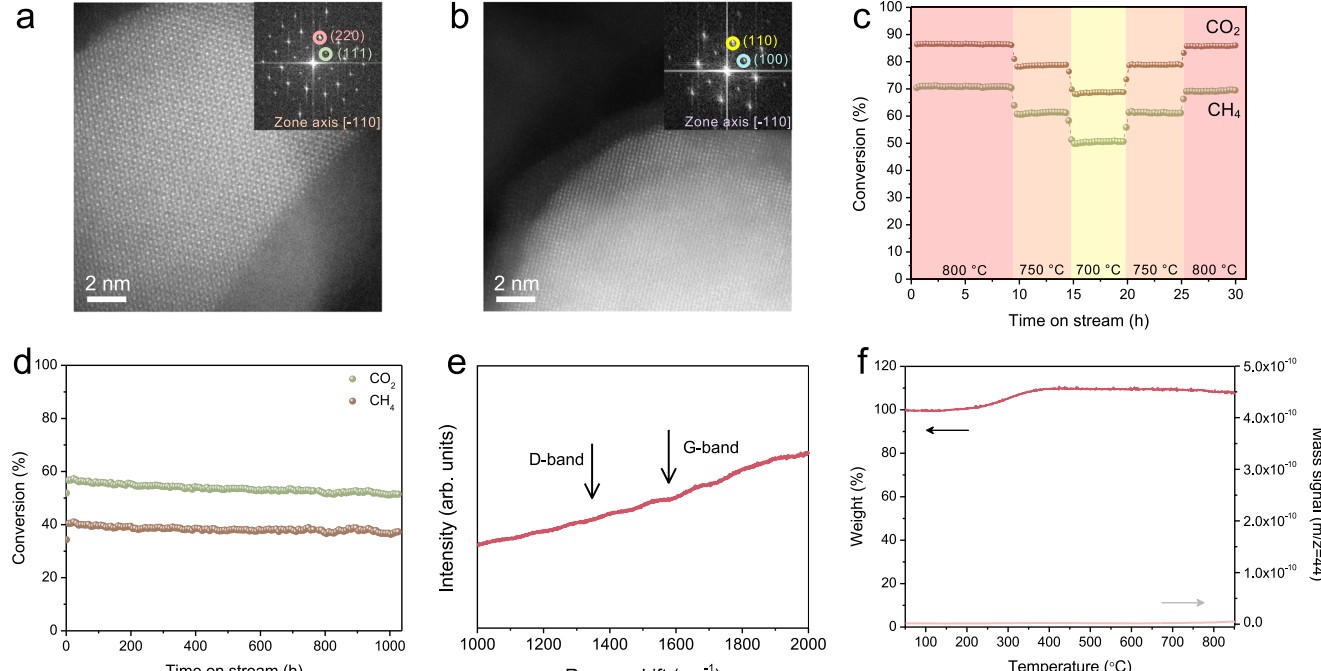

**Fig. 2 | Characterization and dry reforming of methane evaluation of $CoAl_{0.5}Ga_{1.5}O_4$ catalyst.** HR-STEM image of **a** $CoAl_{0.5}Ga_{1.5}O_4$ and **b** reduced $CoAl_{0.5}Ga_{1.5}O_4$; insets: FFT pattern of the HR-STEM image; **c** temperature dependent DRM activity study; **d** 1000 h stability test at 700 °C with 300 $Lg_{cat}^{-1}h^{-1}$ reaction gas feeding. Post-reaction (700 °C, 1000 h) characterization: **e** Raman spectra of the spent catalyst; and **f** TGA of the spent catalyst; red line: TGA, pink line: mass signal at m/z = 44 ($CO_2$).

CoAl$_{0.5}$Ga$_{1.5}$O$_4$-R is easily regenerable with calcination and reduction (Supplementary Fig. 10).

## Experimental mechanism study

To gain insight into the coke resistance mechanism, we performed a series of temperature-programmed surface reactions. It is generally accepted that in DRM reaction, CH$_4$ cracking and the Boudouard reaction (CO disproportionation) are the two main sources of coke formation, while CO$_2$ assists carbon elimination is realized directly via the reverse Boudouard reaction or indirectly via reacting with surface O* generated from CO$_2$ dissociation. Therefore, a CH$_4$-temperature programmed surface reaction (TPSR) experiment followed by CO$_2$-TPSR was performed to test the CH$_4$ activation and CO$_2$ assisted coke removal properties on the spinel oxides derived catalysts. The CH$_4$-TPSR results shown in Fig. 3a, suggest that the Al rich catalysts, the CoAl$_2$O$_4$-R and the CoAl$_{1.5}$Ga$_{0.5}$O$_4$-R catalysts, show two CH$_4$ consumption peaks, while the others have only one CH$_4$ consumption peak with a smaller peak area. Contrastingly, no CH$_4$ consumption was observed on the CoGa$_2$O$_4$-R catalyst. With the increase of Ga content, the temperature required for CH$_4$ decomposition increases. That is, the starting decomposition temperature for the CoAl$_2$O$_4$-R catalyst is 448.3 °C, reaching a maximum at 458.2 °C. For CoAl$_{0.5}$Ga$_{1.5}$O$_4$-R, CH$_4$ consumption starts at 576.9 °C and reaches the maximum at 617.0 °C. Less CH$_4$ consumption and higher temperature indicate that the CoAl$_{0.5}$Ga$_{1.5}$O$_4$-R catalyst has a lower activity for methane activation, thus better coke resistance from CH$_4$ cracking.

CO$_2$-TPSR was performed on the spent catalyst after CH$_4$-TPSR to test the elimination of deposited coke with CO$_2$. Similar to CH$_4$ activation, the CO$_2$ consumption temperature rises with the addition of Ga, but to a much smaller extent, as shown in Fig. 3b, c shows that the peak temperature differences between CH$_4$ activation and CO$_2$-assisted coke elimination decrease from 173.9 °C on CoAl$_2$O$_4$-R to 69.4 °C on CoAl$_{0.5}$Ga$_{1.5}$O$_4$-R. A higher CH$_4$ activation temperature suggests a more difficult coke deposition. With the smallest temperature difference between these two reactions, CoAl$_{0.5}$Ga$_{1.5}$O$_4$-R has a more balanced redox chemistry over the surface, leading to better coke resistance and stability enhancement.

## Computational study

Apart from the unbalanced kinetics between carbon accumulation (via CH$_4$ dehydrogenation) and carbon removal, we also need to balance the carbon removal and the CO$_2$ dissociation process since they are both affected by the oxygen binding energy, i.e., an active O* (weak binding energy) can facilitate coke removal by O* but inhibit the CO$_2$ dissociation process. As a result, we need to understand and control the three processes (CH$_4$ activation, CO$_2$ dissociation, and coke removal). We thus conducted DFT calculations to understand the role of Ga addition in the enhanced coke resistance and provides further insights on CH$_4$ and CO$_2$ activation as well as carbon removal from the

surface. The coke deposition process was analyzed via the dehydrogenation of CH$_4$ to C, the disproportionation of CO (2CO $\rightleftharpoons$ C + CO$_2$), and carbon diffusion; likewise, the coke removal process was evaluated via carbon direct oxidation reaction (C + O $\rightarrow$ CO). The reverse-Boudouard reaction (C + CO$_2$ $\rightleftharpoons$ 2CO) was investigated for CO$_2$ activation and carbon removal (Fig. 4). The CO$_2$ activation via the direct dissociation of CO$_2$ to CO, which supplies O* for coke removal process, was also examined in this study. It is noted that in this study, we focused on the coke formation via deep dehydrogenation of methane[30,31], since its barrier on Co is typically lower than the barrier of C-O dissociation[32,33].

Figure 4b shows the energy diagram of the dehydrogenation of CH$_4$ to C on metallic Co, and intermetallic between Co and Ga with an increasing proportion of Ga in the mixtures from 1:1 (CoGa) to 1:3 (CoGa$_3$). On both surfaces, metallic Co and intermetallic CoGa, the CH$_x$ (x = 1–4) species was first adsorbed on a Co site, followed by C-H bond cleavage to form CH$_{x-1}$* and H* (see Supplementary Fig. 12 for the detailed configuration). However, the intrinsic activation energy of the first C−H bond activation of CH$_4$ decreases from 1.0 to 0.8 eV when Co:Ga ratio is 1:1, and is further reduced to 0.6 eV when this ratio reaches 1:3. By contrast, the barrier for the second C−H bond activation slightly increases. Interestingly, the barrier for the third and fourth C−H bond activation rises to 2.1 eV, when Ga is introduced. Hence, despite promoting the 1$^{st}$ C-H bond activation, introducing Ga suppresses coke formation by inhibiting deep dehydrogenation due to the increase of the barrier of the 3$^{rd}$ and 4$^{th}$ C-H bond cleavage, which is consistent with observed CH$_4$-TPSR in Fig. 3a. Coke can also be formed via the disproportionation of CO. We found that adding Ga can reduce the activation energy of this process by 0.5 eV; however, the barrier for this reaction on CoGa is still as high as 2.5 eV (see Supplementary Fig. 13 and Table 1). Compared to the dehydrogenation process, the disproportionation of CO reaction seems to be less favorable due to its higher endothermic reaction energy and activation barriers. Even if the C atom is deposited on the CoGa surface, the diffusion energy of C atoms on a CoGa surface is much higher than it on a Co surface, implying that carbon cluster formation is inhibited on CoGa (Fig. 4d). The higher diffusion energy of C atoms on CoGa can be attributed to the stronger adsorption energy by 1.4 eV of a C atom on CoGa than on Co. Consequently, the C atoms are trapped at the hollow sites on CoGa (Fig. 4d, labeled as D position). Collectively, these results show that adding Ga can suppress coke formation on the catalyst surface.

The improved coke resistance could also result from an enhanced rate of coke removal. To study this possibility, coke removal was evaluated via the reverse-Boudouard reaction and a carbon direct oxidation reaction. Although the barrier of the former reaction is quite comparable on both metallic Co and its intermetallic CoGa, the barrier of the later reaction is reduced on CoGa by 0.5 eV to 0.9 eV (Fig. 4c, Supplementary Fig. 13). The effectiveness of carbon oxidation of CoGa can be attributed to the weaker binding energy of O* on CoGa than on

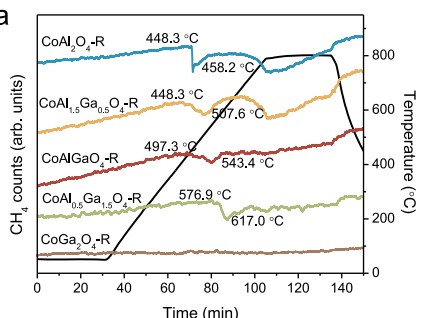
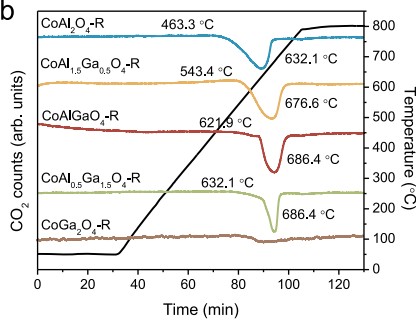
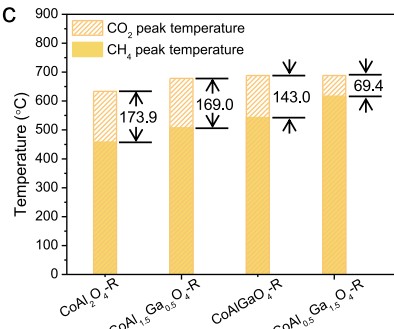

**Fig. 3 | Coke resistance mechanism study. a** CH$_4$-TPSR, mass signal m/z = 15; **b** CO$_2$-TPSR, mass signal m/z = 44, and **c** CH$_4$ and CO$_2$ consumption peak temperatures; peak difference between CO$_2$ and CH$_4$: CoAl$_2$O$_4$-R – 173.9 °C, CoAl$_{1.5}$Ga$_{0.5}$O$_4$-R – 169.0 °C, CoAlGaO$_4$-R – 143.0 °C, and CoAl$_{0.5}$Ga$_{1.5}$O$_4$-R – 69.4 °C.

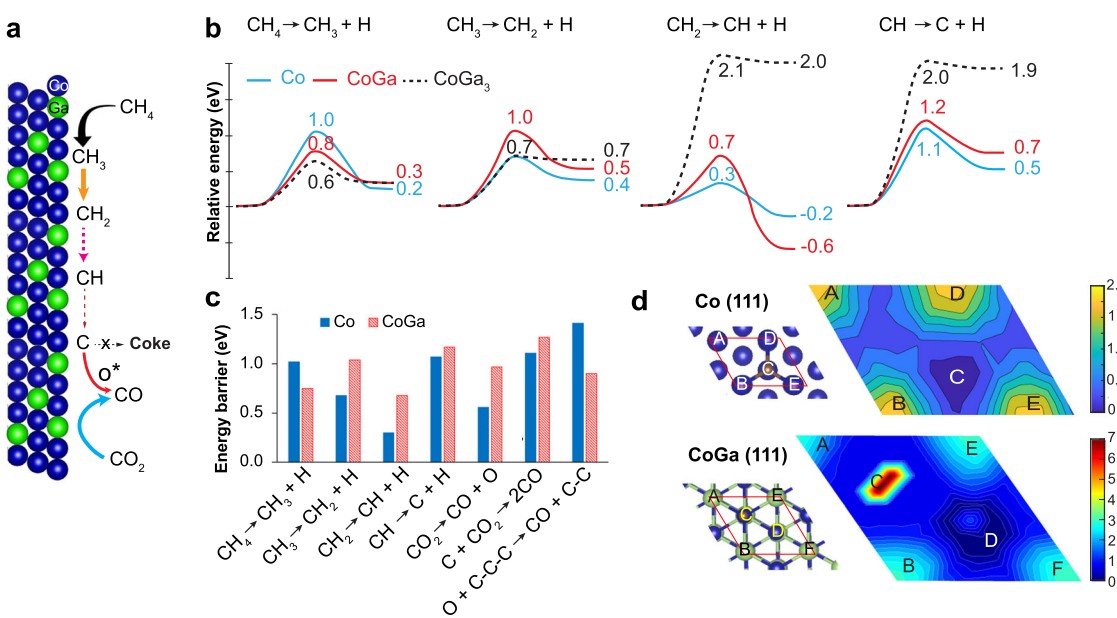

**Fig. 4 | DFT calculations of DRM reactions. a** Schematic illustration of coke resistance on CoGa; **b** Energy diagram of $CH_x$ (x = 1-4) dehydrogenation on Co (blue), CoGa (red), and $CoGa_3$ (black, dash); **c** Summary of the energy barriers on Co and CoGa; **d** Contour plot of relative adsorption energy of C atom on Co (upper panel) and CoGa (lower panel). The level step of both contour plots is 0.25 eV. The unit of color bar is eV. Blue, green, and gray ball are denoted for Co, Ga, and C, respectively.

the Co surface. The DFT-calculated binding energy of an O* adatom on CoGa is weaker by 0.6 eV compared to the Co surface.

We also considered the O-assisted C-H activation in methane and found that the reaction energy between the direct and O-assisted initial C-H dissociation is very similar. For the last C-H bond cleavage, a previous study also showed a similar barrier for both pathways on Co[34]. In addition, the increased barrier to dissociate $CO_2$ on CoGa (Fig. 4c), in line with the higher temperature of $CO_2$-TPSR observed in experiments, should also result in less availability of O* on the surface to support C-H bond cleavage. However, a high coverage of reactive oxygen O* on the catalyst surface is needed to have a significant role for the O*-assisted pathway[35]. Hence, it is expected that the O*-assisted pathway is less pronounced on $CoGa_x$. Therefore, the direct C-H bond cleavage was focused on in this study.

The increased barrier for $CH_4$ deep dehydrogenation to form carbon, suppressed diffusion of surface carbon to aggregate, together with the lower barriers of carbon removal over CoGa highlights the role of Ga and provides an explanation for the slower reaction rates and enhanced catalyst stability. It should be noticed that the computational models were based on well-ordered structures, while the nature and dynamics of the active sites under reaction conditions remain to be explored.

**Generalizability demonstration of the coke-free design principle**

To explore the generalizability of the balancing kinetics for thermodynamics optimization concept, different elemental compositions of spinel oxides were screened. In principle, partially replacing Al from $CoAl_2O_4$ with other 3+ metals ($M^{III}$), such as V, Cr, Mn, and Fe, could yield a coke-free DRM catalyst, $CoAl_xM_yO_4$-R, where the addition of $M^{III}$ could suppress C-H activation to balance with carbon removal. As shown in Supplementary Fig. 14, akin to $CoAl_xGa_{(1-x)}O_4$-R(x ≤ 1), designed $CoAlVO_4$-R kept stable and coke-free in a 100 h DRM test. Although $CoAlMnO_4$-R and $CoAlFeO_4$-R showed decreasing activity over DRM stability tests, no carbon was detected, suggesting a coke-free DRM process (Supplementary Fig. 14). It is worth noting that the deactivation of $CoAlMO_4$-R can be induced by other factors, such as changes in surface structure, which can be further improved by changing Al to $M^{III}$ ratio, modifying pre-treatment conditions, and

tuning DRM reaction conditions to enhance stability while maintaining coke-free. Following the design principle, a series of robust and coke-free catalysts could be rationally designed. With the family of catalysts, the choice of catalyst for the practical application becomes flexible and accommodating catalysts to various requirements becomes feasible.

In summary, this work presents a generalizable and simple synthesis strategy for designing robust and coke-free catalysts by optimizing thermodynamics at the price of chemical kinetics. Dry reforming of methane was studied to illustrate the proof-of-concept. By combining the active and non-active DRM catalysts derived from $CoAl_2O_4$ and $CoGa_2O_4$, a robust DRM catalyst, $CoAl_{0.5}Ga_{1.5}O_4$-R, was discovered which remained stable and coke-free during a 1000 h DRM test. The TPSR experiments and DFT calculations confirmed optimum coke resistance on $CoAl_{0.5}Ga_{1.5}O_4$-R catalyst through compromising the DRM reaction activity. This concept can be applied to other reactions requiring harsh reaction conditions that suffer from severe catalyst deactivation.

# Methods
## Materials
Cobalt(II) nitrate hexahydrate ($Co(NO_3)_2$•$6H_2O$, 99%, Strem Chemicals), aluminum nitrate nonahydrate ($Al(NO_3)_3$•$9H_2O$, 99 + %, Acros Organics), Gallium(III) nitrate hydrate ($Ga(NO_3)_3$•$xH_2O$, 99.9998% trace metal basis, Acros Organics), Iron(III) nitrate nonahydrate ($Fe(NO_3)_3$•$9H_2O$, ≥98% ACS reagent, Sigma Aldrich), Vanadium chloride ($VCl_3$, purified, City Chemical Cooperation), Manganese (II) chloride ($MnCl_2$, 97%, Acros Organics), and ammonium hydroxide ($NH_4OH$, 28-30%, Fisher Chemical) were used as received. Milli-Q water (18.2 MΩ·cm at 25 °C) was used during all the experimental procedures, including glassware cleaning and sample preparation.

## Synthesis of Co related spinel oxides
The spinel oxide was synthesized using a reported method with modification[36]. Typically, in a 125 mL polypropene bottle, 3 mmol $A(NO_3)_2$ and 6 mmol $[B(NO_3)_3 + C(NO_3)_3]$ were dissolved in 9 mL water. The solution pH was adjusted to 10 using diluted ammonia (~ 3% ammonium hydroxide aqueous solution) added dropwise under continuous stirring. The clear solution became cloudy during the

**Table 1 | The amount of metal nitrate precursors used for the synthesis of different spinel oxides $AB_xC_{2-x}O_4$**

| | $A(NO_3)_2$ | $B(NO_3)_3$ | $C(NO_3)_3$ |
|---|---|---|---|
| Spinel oxide | $Co(NO_3)_2 \cdot 6H_2O$ | $Al(NO_3)_3 \cdot 9H_2O$ | $Ga(NO_3)_3 \cdot xH_2O$ |
| $CoAl_2O_4$ | 3 mmol, 873.09 mg | 6 mmol, 2250.78 mg | 0 |
| $CoAl_{1.5}Ga_{0.5}O_4$ | 3 mmol, 873.09 mg | 4.5 mmol, 1688.08 mg | 1.5 mmol, 383.61 mg |
| $CoAlGaO_4$ | 3 mmol, 873.09 mg | 3 mmol, 1125.39 mg | 3 mmol, 767.22 mg |
| $CoAl_{0.5}Ga_{1.5}O_4$ | 3 mmol, 873.09 mg | 1.5 mmol, 562.69 mg | 4.5 mmol, 1150.83 mg |
| $CoGa_2O_4$ | 3 mmol, 873.09 mg | 0 | 6 mmol, 1534.44 mg |
| $CoAlFeO_4$ | 3 mmol, 873.09 mg | 3 mmol, 1125.39 mg | $Fe(NO_3)_3 \cdot 9H_2O$ 3 mmol, 1212.00 mg |
| $CoAlVO_4$ | 3 mmol, 873.09 mg | 3 mmol, 1125.39 mg | $VCl_3$ 3 mmol, 471.90 mg |
| $CoAlMnO_4$ | 3 mmol, 873.09 mg | 3 mmol, 1125.39 mg | $MnCl_2$ 3 mmol, 377.52 mg |

addition of the diluted ammonia, indicating the formation of metal hydroxides. After that, the mixture was stirred at 600 rpm for 24 h at room temperature. The precipitate was centrifuged at 8000 rpm (8157 × g) for 3 min, washed with water three times, and dried in 100 °C oven for 12 h. After drying, the precipitate changed from gel to hard solid chunks. The solid chunks were grounded to fine powder before calcining at 800 °C for 2 h with a ramping rate of 2 °C min$^{-1}$ from 25 to 800 °C. After calcination, the obtained spinel oxides were named as $AB_xC_{2-x}O_4$, where x equals the mmol of $B(NO_3)_3$ divided by 3 (Table 1). The prepared spinel oxides catalysts were stored in air before use.

## Characterizations

TEM samples were prepared through drop casting method on Cu grids. HAADF-STEM imaging was performed on a Titan Themis 300 equipped with gun monochromator and probe spherical aberration (Cs) corrector operated at 200 kV. The EDS mapping data was collected with a Super-X EDX detector. Powder X-ray diffraction (PXRD) was performed on a Bruker D8 Advance Twin diffractometer using Cu Kα1 radiation (40 kV, 40 mA, λ = 0.1541 nm). SEM samples were prepared through drop casting catalyst-ethanol suspension on a silicon wafer. SEM was performed on JEOL JSM-IT200 Scanning Electron Microscope equipped with EDS detector. SEM image of after 1000-h test was obtained on JEOL JSM-7900FLV Scanning Elelctron Microscope. The XPS measurements were performed using a Kratos Amicus/ESCA 3400 instrument. The sample was irradiated with 240 W unmonochromated Al x-rays, and photoelectrons emitted at 0° from the surface normal were energy analyzed using a DuPont type analyzer. The pass energy was set at 150 eV. CasaXPS was used to process raw data. Symmetric line shapes with equal peak width were used in the peak fitting. X-ray absorption spectra (XAS) at Ga K-edge and Co K-edge were recorded at the Inner Shell Spectroscopy (ISS, 8-ID) beamline of the National Synchrotron Light Source II at Brookhaven National Laboratory. Using a Si (111) double-crystal monochromator. Bare Si flat mirror placed upstream of the sample reduced the contribution from high harmonics. Spectra were recorded in transmission using ion chambers with 35%/65% N$_2$/He mixture. Beam size at the sample was ~800 μm in diameter. The photon flux was estimated to be 2 × 10$^{13}$ ph/s. Near-edge X-ray absorption spectroscopy (XANES) and Extended X-ray Absorption Fine Structure (EXAFS) data were collected at room temperature, with energy calibrated using a Co and W foils. XAS data were normalized and Fourier transformed using Athena software[37].

## Dry reforming of methane

**Pretreatment.** Catalysts were reduced at 750 °C for 2 h under the flow of 20% H$_2$/He (H$_2$:He=1:4, 100 standard cubic centimeters per minute, SCCM). After 2 h reduction, He gas (100 SCCM) was flowed through the whole system for 1 hour. Meanwhile, the temperature was slowly increased to 800 °C or decreased to 700 °C in 25 min.

**Catalyst evaluation.** Dry reforming of methane tests were conducted in a home build catalysts evaluation system. 20 mg as-synthesized spinel oxides were mixed with quartz sand to 300 mg. Catalysts were loaded in a U-shape reactor (inner diameter of 8 mm) with extra 500 mg quartz sand and quartz wool on the top and bottom of catalysts. The products of DRM were detected by online gas chromatography (GC, Agilent 5890) equipped with a TCD detector. A Carboxen 1000 (15 ft × 1/8 in × 2.1 mm) column was chosen to separate the gas mixture. Using the difference in thermal conductivity between the carrier gas and the target gas, CH$_4$, CO$_2$, and CO was detected with He gas as the GC carrier gas. The temperature of the catalyst bed was controlled and recorded by a temperature controller with a K-type thermocouple. All the reactant gases, CO$_2$, CH$_4$, and He, are in ultra-high purity grade and used without further purification or special treatment. The DRM performance was tested under 800 °C to demonstrate high activity capability with total gas-hourly-space-velocity (GHSV) of 300 $Lg_{cat}^{-1}h^{-1}$, CH$_4$:CO$_2$:He = 10:10:80 SCCM (Supplementary Fig. 2). The stability tests were performance under 700 °C with the same gas flow.

Stability tests were also performed on catalysts under the same flow and temperature conditions but without quartz sand dilution for the post-reaction characterization. 20 mg spinel oxide was loaded on the U-tube reactor with quartz wool and a quartz filter on the bottom. Post-reaction catalyst can be regenerated through 2 h calcination at 800 °C under air followed by 2 h reduction at 750 °C under 20% H$_2$/He.

## Post DRM reaction characterizations

**Oxygen temperature-programmed oxidation (O$_2$-TPO).** O$_2$-TPO of the post reaction catalysts was tested in the same gas phase reactor system with mass spectroscopy (Agilent 5973 Inert Mass Selective Detector, MSD) connected to the reactor outlet. Before the O$_2$-TPO test, the catalysts were treated with 40 SCCM He at 150 °C for 1 h to remove the adsorbed gases. After cooling to 50 °C, 3% O$_2$ was induced (O$_2$:He=1.2:38.8 SCCM). Once the mass signal stayed stable, the catalysts were heated to 900 °C with a ramping rate of 10 °C min$^{-1}$. The mass signal of H$_2$O (m/z = 18), CO (m/z = 28), O$_2$ (m/z = 32), and CO$_2$ (m/z = 44) were recorded.

**Thermogravimetric analysis (TGA).** TGA was performed on Netzsch STA499 F1 equipped with a mass spectroscopy detector. Samples were heated to 850 °C with 10 °C min$^{-1}$ under air flow of 40 mL min$^{-1}$. The mass signal of CO$_2$ (m/z = 44) was collected.

**Raman spectroscopy.** Raman spectra were obtained on Horiba Jobin Yvon iHR 550 imaging spectrometer, with laser wavelength of 532 nm. Raman spectra of 1000 h stability test sample were obtained on Horiba LabRAM HR Evolution Confocal RAMAN. All the Raman spectra were directly exported and plotted without any post-measurement data processing.

## Temperature programmed reaction

**Methane temperature-programmed surface reaction (CH₄-TPSR).**
50 mg spinel oxides mixed with quartz sand were packed in a U-tube reactor connecting to an MSD. The spinel oxides were pre-reduced at 750 °C for 2 h under 20% $H_2$ in He (20 SCCM $H_2$, 80 SCCM He) and cooled down to 50 °C under the flow of He (100 SCCM). 3% $CH_4$ in He ($CH_4$:He=1.2:38.8 SCCM) were flowed through the reactor at 50 °C. After the MS signal of $CH_4$ (m/z = 15) reached a stable level, the temperature was ramped to 900 °C at a rate of 10 °C min⁻¹.

**CO₂-TPSR.** $CO_2$-TPSR was performed on the catalysts after $CH_4$-TPSR. After $CH_4$-TPSR, the reactor was cooled down to 50 °C under flow of He (40 SCCM). After the $CO_2$ MS signal stabilized under the flow of 40 SCCM 3% $CO_2$/He at 50 °C ($CO_2$:He=1.2:38.8 SCCM), the temperature was ramped to 900 °C at a rate of 10 °C min⁻¹.

## Computational details

DFT calculations were performed using the VASP package[38]. The exchange-correlation functional PBE was used[39]. The van der Waals interaction was included by employing the DFT-D3 dispersion model[40]. The model for Co was constructed based on four layers of (3 × 3) repeated slabs of Co (111) with a lattice parameter of a = 3.48 Å taken from the optimization of the FCC Co unit cell. While the top two layers were fully relaxed, the bottom three layers were fixed at their bulk positions. The model for CoGa was constructed based (3 × 3) repeated slabs of CoGa (111) with a lattice parameter of a = 2.85 Å taken from the optimization of the CoGa unit cell. The Co-terminated CoGa (111) surface structure had six and a half Co-Ga layers in which the bottom four layers were fixed at their bulk position while the top two and a half layer were fully relaxed. The model for CoGa₃ was constructed based on P4n2 CoGa₃ unit cell with optimized lattice parameter of a = 6.20 Å. The 2 × 2 CoGa₃ (111) with 2 CoGa₃ layers (the top layer was fully relaxed while the bottom layer was fixed as their bulk position) were used to model the CoGa₃ catalyst. A vacuum of 15 Å was added along the z axis perpendicular to the surface for all models (see Supplementary Fig. 12 for the detail model of catalyst surface). The energy convergence for electronic relaxation was set to 10⁻⁴ eV. All atoms were relaxed until the atomic force <0.02 eV/Å with a kinetic energy cutoff of 500 eV. The total energy calculations reported in this study used a k-point grid of (1×1×1). We compared the results with a (3 × 3 × 1) calculation and found that the activation energy and reaction energy differed by 0.1 to 0.2 eV between these two settings of the k-points for CoGa and Co. Therefore, the results obtained using a k-point grid of (1 × 1 × 1) were reported in this study to reduce the computational cost. The nudged elastic band (NEB) method and the dimer method were used to find the transition state and calculate the activation energy barriers[41,42]. The optimized configurations can be found in Supplementary Data 1. The spin polarization is considered in all the calculations of Co (111). The calculated magnetic moment of Co is 1.64 μB/atom, which is quite close to the experimental value (1.71 μB/atom)[34]. The NEB calculation of Co (111) was conducted based on the optimized configurations without spin (except the NEB calculation of $CO_2$ dissociation). These structures were used for the spin-polarized calculations without further relaxation. We checked and found fully relaxing in spin-polarized calculations would lead to a change around 0.1 eV for the reaction energy and almost zero for the activation energy.

The adsorption energy is defined by:

$$\Delta E_{ads} = E_{adsorbate/surface} - E_{surface} - E_{adsorbate} \qquad (1)$$

in which the adsorption energy of C atom and O atom on the surface was referred to energy of C in graphite and O in oxygen molecule respectively.

All optimized configurations obtained via DFT calculations are available in Supplementary Data 1.

## Data availability

All data are available in the main text or the supplementary materials.

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

## Acknowledgements

J.Y. and W.H. thank the support from NSF (CHE-2108306/2108307) and Trapp Innovation Award. T.L. and B.W. were supported by the Department of Energy, Basic Energy Sciences (Grant No. DE-SC0020300). The DFT calculations were performed using computational resources at the OU Supercomputing Center for Education & Research (OSCER) at the University of Oklahoma and the National Energy Research Scientific Computing Center (NERSC), a U.S. Department of Energy Office of Science User Facility. This research used ISS (8-ID) beamline of the National Synchrotron Light Source II, a U.S. Department of Energy (DOE) Office of Science User Facility operated for the DOE Office of Science by Brookhaven National Laboratory under Contract No. DE-SC0012704. We thank Dr. Bruce Ravel for providing reference XAS spectra. This work made use of the EPIC and SPID facility of Northwestern University's NUANCE Center, which has received support from the SHyNE Resource (NSF ECCS-2025633), the IIN, and Northwestern's MRSEC program (NSF DMR-1720139). The authors acknowledge Dr. Xinwei Wang for his valuable discussions on Raman characterization.

## Author contributions

J.Y. conducted catalysts synthesis, characterization, performance evaluation, and experimental mechanism study; T.L. performed theoretical study; D.J. did XPS measurement and analysis; E.S. and D.L. performed XAS measurement and data analysis; N.H. did the Raman characterization; K.L. helped post DRM catalyst characterizations; D.E.R. provided suggestions on the catalysts' evaluation; E.H.S. supervised the manuscript construction; B.W. supervised theoretical study; and W.H. supervised the experimental study. W.H., B.W., and E.H.S. conceived and managed the overall project. All authors contributed to writing the manuscript.

## Competing interests

The authors declare no competing interests.
