## [Peer Review File · Nature Communications]

REVIEWER COMMENTS

Reviewer #1 (Remarks to the Author):

Huang and co-workers reported the coking-resistant DRM over the rationally designed CoAlGa catalysts, where sufficient performances and insight into the extraordinary catalytic performances have been achieved and discussed. The results are interesting, and I strongly support its publication in Nature Communications. The following issues should be considered during the revision.

1. I fully understood the authors' concern about employing the reaction conditions, where the diluted CH₄/CO₂ gases and high GHSV were employed to avoid the equilibrium conversion. However, for a catalyst with potential for industrial application, as mentioned in the manuscript, the persuasive evaluation should be performed using undiluted reactants and sufficient GHSV, which should be added in the main text of the revised manuscript.
2. The fine structure of the spent catalyst (after reaction or even after regeneration) was necessary to confirm the maintained structure during the harsh reaction conditions.
3. For the coke formation channel, the C-O dissociation of CO and deep dehydrogenation of methane have been reported as crucial steps. These channels have been systemically studied previously, where some important references should be included.
4. How about the comparison with the efficient nickel catalysts tested previously? A table summarizing the catalytic properties of this Co catalyst and the previous nickel catalyst would benefit easily understanding the features of new catalysts.

Reviewer #2 (Remarks to the Author):

The authors reported on a spinel-oxide-derived cobalt catalyst that appeared to be quite stable over 1000 h in the dry reforming of methane at the tested conditions. This may be an important discovery, but I could not recommend its publication in its present state, because of the numerous flaws contained in the manuscript (points 1-5 are major critiques). My overall impression is that the so-called BKTO concept carries little weight in terms of its scientific essence, since the mechanism of

deactivation and surface reactions that lead to catalytically consequential carbon deposits remain poorly elucidated.

(1) Throughout the paper, the authors claimed that unbalanced kinetics between “fast C-H activation and sluggish CO₂ activation” cause rapid deactivation. However, this is categorically wrong, because it is well established that C-H bond activation typically limits the rate of methane reforming, and CO₂ activation is much more rapid and often is quasi-equilibrated. The cited paper on one occasion (ref6), which was intended to support their erroneous claim, actually contradicts the statement/opinion of the present authors’. A relevant excerpt of ref6 is provided below:

“Mechanistically the most important and the slowest step in DRM is the activation of CH₄ which occurs primarily on the metallic site and thus having high dispersion of the metal is important. Activation of CO₂ is relatively a faster process and occurs mainly on the support or the metal–support interface in case of acidic and basic supports.” In fact, one can find it more or less a consensus in the literature that CO₂ activation is NOT sluggish and C-H activation is NOT fast relative to CO₂ activation (the works of Iglesia et al.), not only for noble metals but also for Ni and Co (e.g., J. Am. Chem. Soc. 2017, 139, 20, 6928–6945). What is really “sluggish” (or perhaps only in way that does not quite keep up with the deposition rate of surface C*) and the reason for carbon accumulation is the less rapid removal of C* by the oxidizing species on the surface, but NOT because of a “sluggish” CO₂ activation. It is really disappointing to see such conceptual mistakes/layman’s view from a group of reputed authors. If the present authors disagree, then please provide counter-evidence that would speak against what is reasoned above, and evidence that would support their opinion and their perception/reading of the current literature that CO₂ activation is sluggish and C-H activation is relatively fast. I would also request to see such evidence for their own catalysts, including isotopic scrambling and isotopic effects as nicely exploited as mechanistic proof in the previous works of Iglesia and others.

(2) The authors appear to have equated CO₂ activation with a direct reaction between CO₂ and C* on the surface. This is also conceptually inappropriate because the direct reaction between CO₂ and the carbon deposits is not supported by any mainstream studies as catalytically relevant for methane reforming on metal catalysts. CO₂ activation typically proceeds on vacant metal sites, and the removal of C* is assisted by O* or OH* species derived from the weak or strong oxidant (CO₂, H₂O, O₂) in representative sets of reforming chemistry (DRM, steam reforming, tri-reforming, or autothermal reforming). Accordingly, SuppleFig. 1 requires to be corrected.

(3) There does not seem to be a significant difference in the catalytic stability among the CoAl spinel oxide catalysts shown in Fig. 1. For the CoAl₂O₄ catalyst, the deactivation mainly occurred at the initial 15 h, which could be due to some acid-catalyzed carbon deposition on the support, i.e., not necessarily associated with coke deposition on metal particles and the catalyst stability. The authors need to check the carbon balance before and after this initial stage.

(4) The 1000 h test is good to have and the absence of coke deposits may be a remarkable finding. From the Raman spectra and the TPO profile in Figure S4a, it can be indeed confirmed that there is no carbon deposition on the CoAlGaO₄ catalyst. However, this catalyst exhibited a similar stability as the CoAl₂O₄ catalyst, or perhaps somewhat inferior (Fig. 1). The additional CoAlMnO₄-R and CoAlFeO₄-R catalysts also showed visible deactivation, without detectable coke deposition. Thus, given the lack of a compelling relationship between the stability data and TPO/Raman

characterization results, I'm left wondering what really is the decisive evidence that links the measured carbon deposits to the observed stability?

(5) DFT calculations: "These increases in the barrier of the later C-H bond activation suggest that adding Ga could suppress coke formation by inhibiting deep dehydrogenation, which is consistent with observed CH₄-TPSR in Fig. 3a." By "increases in the barrier of the later C-H bond activation", do the authors mean that the formation of *CH₂O before the cleavage of C-H of *CH₂? Then, what is the barrier energy for the cleavage of C-H of the formed *CH₂O and the following *CHO? Moreover, from the DFT-computed energy barriers, the introduction of Ga did not suppress but promote the activation of methane (the first C-H bond activation is typically rate-limiting), while the activation of CO₂ or reverse Boudouard reaction was suppressed (Fig. 4c). I'm afraid I cannot see any consistency here.

(6) This work contains a few inappropriate expressions. For example, "... have approached good activities reaching the equilibrium reaction rate...". What on earth is "equilibrium reaction rate"? At equilibrium, net reaction rate is zero and there is no way to determine the forward and reverse rate. Therefore, there is no such thing that can be called an "equilibrium reaction rate".

(7) The peaks for the Ga-rich samples seem too small to be identified as meaningful information (Fig. 3).

(8) The results of CO₂-TPSR cannot be adequately explained without knowing the manner of CO₂ activation. From your DFT calculation results (Figure S11), the direct dissociation of CO₂ to CO and C can better explain the CO₂-TPSR data, but not the reverse Boudouard reaction.

(9) This reviewer is unsure whether CoAl₂O₄ could be said to be "a previously well-studied active DRM catalyst". The authors may want to provide some good citations that would support this claim. To prove that it is really a catalyst with good activity, the authors should calculate the reaction rate (which should be the forward reaction rate, not the net reaction rate), rather than conversion of methane. Then it would be better to put such an activity into perspective, i.e., comparing with other catalysts reported.

Response to Reviewers' Comments:

We appreciate the reviewers' constructive comments on our manuscript. We have revised the manuscript according to these comments and responded to all questions. The original questions are colored **black**, our answers are in **blue**, and the revisions to the manuscript are in **red**.

Reviewer #1 (Remarks to the Author):

Huang and co-workers reported the coking-resistant DRM over the rationally designed CoAlGa catalysts, where sufficient performances and insight into the extraordinary catalytic performances have been achieved and discussed. The results are interesting, and I strongly support its publication in Nature Communications. The following issues should be considered during the revision.

1. I fully understood the authors' concern about employing the reaction conditions, where the diluted CH₄/CO₂ gases and high GHSV were employed to avoid the equilibrium conversion. However, for a catalyst with potential for industrial application, as mentioned in the manuscript, the persuasive evaluation should be performed using undiluted reactants and sufficient GHSV, which should be added in the main text of the revised manuscript.

Reply: We conducted a non-diluted DRM experiment using the CoAl_{0.5}Ga_{1.5}O₄-R catalyst at the same GHSV (300 L g_{cat}⁻¹ h⁻¹) as diluted gas. Despite demonstrating stable performance over a 20-hour test, we observed coke deposition (see Figure below and now Supplementary Figure S11). We attribute the formation of coke to the altered kinetics resulting from changes in the partial pressure of the reactants. The field of DRM has been suffering from coke depositions for decades. The coke-free catalysts have been very rarely discovered even with diluted gases. Based on our extensive literature survey, maintaining catalyst stability under pure DRM conditions is highly challenging, and there are no reported demonstrations of coke-free DRM using non-diluted CH₄ and CO₂. However, the work that we report here lays a foundation for addressing this challenge: through further efforts in altering catalysis composition to compensate for the kinetics under the pure stream, a coke-free DRM process is potentially achievable by applying our design concept shown in this work.

We added the pure DRM stream test result to supplementary information.

Supplementary Fig. 11 | DRM evaluation of CoAl_{0.5}Ga_{1.5}O₄-R with non-diluted CH₄ and CO₂ gas mixtures. Reaction conditions: 700 °C, CH₄/CO₂=50/50 SCCM, GHSV = 300 L g_{cat}⁻¹ h⁻¹.

2. The fine structure of the spent catalyst (after reaction or even after regeneration) was necessary to confirm the maintained structure during the harsh reaction conditions.

Reply: We performed detailed structural characterization to test the re-generatability of the catalyst. As shown in the Supplementary Fig. 10, after the 1000-h stability test, the catalyst was calcined under 800 °C for 2h and reduced under 750 °C for another 2h. The catalyst after both calcination and reduction shows the same structure as the original catalyst, and the CoGa intermetallic structure is preserved.

We added the following figure and analysis to the supplementary information and the following sentence on Page 5 of the manuscript, “CoAl_{0.5}Ga_{1.5}O₄-R is easily regenerable with calcination and reduction (Supplementary Fig. 10).”, and to the methods section, “Post-reaction catalyst can be regenerated through 2 h calcination at 800 °C under air followed by 2 h reduction at 750 °C under 20% H₂/He.”

We added regeneration procedure description to the method section under catalyst evaluation sub-section.

Supplementary Fig. 10 | Regeneration characterizations. (a) PXRD pattern of regenerated $\text{CoAl}_{0.5}\text{Ga}_{1.5}\text{O}_4$ (pink) and regenerated $\text{CoAl}_{0.5}\text{Ga}_{1.5}\text{O}_4\text{-R}$ (yellow), compared with original $\text{CoAl}_{0.5}\text{Ga}_{1.5}\text{O}_4\text{-R}$ (green), and $\text{CoAl}_{0.5}\text{Ga}_{1.5}\text{O}_4$ (blue). The XRD patterns of regenerated $\text{CoAl}_{0.5}\text{Ga}_{1.5}\text{O}_4$ and regenerated $\text{CoAl}_{0.5}\text{Ga}_{1.5}\text{O}_4\text{-R}$ match with original $\text{CoAl}_{0.5}\text{Ga}_{1.5}\text{O}_4$ and $\text{CoAl}_{0.5}\text{Ga}_{1.5}\text{O}_4\text{-R}$, respectively. (b) HR-STEM image of regenerated $\text{CoAl}_{0.5}\text{Ga}_{1.5}\text{O}_4\text{-R}$; inset FFT of (b), showing CoGa intermetallic structure. (c) STEM-EDS mapping of regenerated $\text{CoAl}_{0.5}\text{Ga}_{1.5}\text{O}_4\text{-R}$. Analysis of the images reveals that the designed $\text{CoAl}_{0.5}\text{Ga}_{1.5}\text{O}_4\text{-R}$ is regenerable with simple calcination and reduction.

3. For the coke formation channel, the C-O dissociation of CO and deep dehydrogenation of methane have been reported as crucial steps. These channels have been systemically studied previously, where some important references should be included.

Reply: We thank the reviewer for the valuable suggestion. Both mechanisms have been proposed for coke formation in literature. In our study, we focused on the coke formation via deep dehydrogenation of methane since its barrier on Co is typically lower than the barrier of C-O dissociation.

We added the following references to acknowledge previous DRM mechanism study.

References of the deep dehydrogenation of methane:

- 32 Wang, Y., Hu, P., Yang, J., Zhu, Y. A. & Chen. C-H bond activation in light alkanes: a theoretical perspective. *Chemical Society reviews* **50**, 4299-4358 (2021). <https://doi.org/10.1039/d0cs01262a>
- 33 Zhu Q, *et al.* Zeolite fixed cobalt–nickel nanoparticles for coking and sintering resistance in dry reforming of methane. *Chem Eng Sci* **280**, 119030 (2023). <https://doi.org/10.1016/j.ces.2023.119030>

References of the C-O dissociation to form C and O on Co surface:

- 34 Liu, J.-X., Su, H.-Y., Sun, D.-P., Zhang, B.-Y. & Li, W.-X. Crystallographic Dependence of CO Activation on Cobalt Catalysts: HCP versus FCC. *J. Am. Chem. Soc.* **135**, 16284-16287 (2013). <https://doi.org/10.1021/ja408521w>

We added the following sentences on Page 7 of the manuscript to the computational study part to clarify it.

“The CO₂ activation via the direct dissociation of CO₂ to CO, which supplies O* for coke removal process, was also examined in this study. It is noted that in this study, we focused on the coke formation via deep dehydrogenation of methane^{32,33}, since its barrier on Co is typically lower than the barrier of C-O dissociation^{34,35}”

4. How about the comparison with the efficient nickel catalysts tested previously? A table summarizing the catalytic properties of this Co catalyst and the previous nickel catalyst would benefit easily understanding the features of new catalysts.

Reply: We added this table to the supporting information to compare the catalyst performance from this work with the reported state-of-the-art DRM catalysts.

On Page 5 of the manuscript, we modified the sentence, “The over 1000 h stability test and the post characterization of the used catalyst conclude that the CoAl_{0.5}Ga_{1.5}O₄-R is a robust and coke-free DRM catalyst with potential for industrial-scale application (stability comparison of CoAl_{0.5}Ga_{1.5}O₄-R and state-of-the-art DRM catalysts shown in Supplementary Table 2)”.

Supplementary Table 2 | Performance comparison of DRM catalysts.

Catalyst	Flow	Temp. (°C)	Rate ($mol\ g_{cat}^{-1}\ h^{-1}$)	Coke formation	Reference
CoAl_{0.5}Ga_{1.5}O₄-R	300 L $g_{cat}^{-1}\ h^{-1}$ CH₄:CO₂:He=1:1:8	700	CH₄: 0.51 CO₂: 0.70	1000 h coke-free	This work
Mo doped Ni on single crystal MgO ^b	60 L $g_{cat}^{-1}\ h^{-1}$ CH ₄ :CO ₂ :He=1:1:8	800	CH ₄ : ~0.27 CO ₂ : ~0.27	850 h coke-free	Science 367 , 777-781 (2020)
Ni atomically dispersed over CeOx doped hydroxyapatite	60 L $g_{cat}^{-1}\ h^{-1}$ CH ₄ :CO ₂ :He=1:1:3	750	CH ₄ : 196.4 CO ₂ : 330.1	100 h ~2% weight loss by TGA	Nat. Commun. 10 , 5181 (2019)
Ni@BOx/h-BN	60 L $g_{cat}^{-1}\ h^{-1}$ CH ₄ :CO ₂ :N ₂ =2:2:1	750	CH ₄ : 0.78 CO ₂ : 0.86	40 h No coke from TEM&SEM	J. Am. Chem. Soc. 142 , 17167–17174 (2020)
Multielement oxide layer confined Ni	30 L $g_{cat}^{-1}\ h^{-1}$ CH ₄ :CO ₂ :N ₂ =1:1:3	800	CH ₄ : ~0.19 to 0.21 CO ₂ : ~0.20 to 0.22	300 h coke-free	ACS Catal. 11 , 12409–12416 (2021)

Ni/ZrO ₂ @BN	25 L $g_{cat}^{-1} h^{-1}$ CH ₄ :CO ₂ =1:1	750	CH ₄ : ~0.39 CO ₂ : ~0.45	200 h 2.8 wt.% coke	Appl. Catal. B 302 , 120859 (2022)
Ni/Ce _{0.9} Eu _{0.1} O _{1.95}	60 L $g_{cat}^{-1} h^{-1}$ CH ₄ :CO ₂ :N ₂ =1:1:2	600	CH ₄ : 0.16 CO ₂ : 0.23	700 min 18.7% weight loss from TGA	J. Catal. 407 , 77–89 (2022)
Ni@HZSM-5	690 L $g_{cat}^{-1} h^{-1}$ 33.0%CO ₂ /10.6%CH ₄ /2.6% Ar/53.8% He	500	CO formation: 0.348 $mol_{CO} g_{Ni}^{-1} h^{-1}$	20 h ~0.43 wt.% coke	Nat. Catal. 5 , 1030-1037 (2022)

Reviewer #2 (Remarks to the Author):

The authors reported on a spinel-oxide-derived cobalt catalyst that appeared to be quite stable over 1000 h in the dry reforming of methane at the tested conditions. This may be an important discovery, but I could not recommend its publication in its present state, because of the numerous flaws contained in the manuscript (points 1-5 are major critiques). My overall impression is that the so-called BKTO concept carries little weight in terms of its scientific essence, since the mechanism of deactivation and surface reactions that lead to catalytically consequential carbon deposits remain poorly elucidated.

(1) Throughout the paper, the authors claimed that unbalanced kinetics between “fast C-H activation and sluggish CO₂ activation” cause rapid deactivation. However, this is categorically wrong, because it is well established that C-H bond activation typically limits the rate of methane reforming, and CO₂ activation is much more rapid and often is quasi-equilibrated.

The cited paper on one occasion (ref6), which was intended to support their erroneous claim, actually contradicts the statement/opinion of the present authors’.

A relevant excerpt of ref6 is provided below: “Mechanistically the most important and the slowest step in DRM is the activation of CH₄ which occurs primarily on the metallic site and thus having high dispersion of the metal is important. Activation of CO₂ is relatively a faster process and occurs mainly on the support or the metal–support interface in case of acidic and basic supports.”

In fact, one can find it more or less a consensus in the literature that CO₂ activation is NOT sluggish and C-H activation is NOT fast relative to CO₂ activation (the works of Iglesia et al.), not only for noble metals but also for Ni and Co (e.g., *J. Am. Chem. Soc.* 2017, 139, 20, 6928–6945). What is really “sluggish” (or perhaps only in way that does not quite keep up with the deposition rate of surface C*) and the reason for carbon accumulation is the less rapid removal of C* by the oxidizing species on the surface, but NOT because of a “sluggish” CO₂ activation.

It is really disappointing to see such conceptual mistakes/layman's view from a group of reputed authors. If the present authors disagree, then please provide counter-evidence that would speak against what is reasoned above, and evidence that would support their opinion and their perception/reading of the current literature that CO₂ activation is sluggish and C-H activation is relatively fast.

I would also request to see such evidence for their own catalysts, including isotopic scrambling and isotopic effects as nicely exploited as mechanistic proof in the previous works of Iglesia and others.

Reply: We appreciate the reviewer's valuable comments. We meant to refer to the unbalance between CH₄ activation and carbon removal. We changed the related misunderstanding expression accordingly throughout the whole manuscript.

We agree with the reviewer that CO₂ activation is usually easier compared to CH₄ dehydrogenation. Our DFT calculations on Co have also shown that the activation barrier of CO₂ direct dissociation on Co is lower than CH₄ dehydrogenation; these two become comparable over CoGa. The barrier of carbon removal is higher than the barrier CH₄ activation over Co, which could lead to the C* accumulation on the Co surface once CH₄ is dehydrogenated. So, the unbalanced kinetics between C accumulation (via CH₄ dehydrogenation) and C removal cause the coke formation. In addition, the binding energy of O* can affect both the CO₂ dissociation and the coke removal process. The weaker binding energy of O* on CoGa_x than it over the pure Co, leads to a higher barrier of CO₂ dissociation but a lower barrier for removing coke by O*.

We believe, with the corrected and more accurate language, that the core catalyst design principle in this work—balancing kinetics of elementary steps (illustrated in Supplementary Fig. 1 below) for the overall thermodynamics optimization—is impactful in the catalyst community.

We revised the manuscript accordingly:

In Abstract, we changed “Dry reforming of methane, a process operated at high temperature, usually involves fast C-H activation but sluggish CO₂ activation, resulting in coke deposition and catalyst deactivation.” to “Dry reforming of methane, a process operated at high temperature, usually involves fast C-H activation but sluggish carbon removal, resulting in coke deposition and catalyst deactivation”.

In the second paragraph of introduction, we changed “With numerous efforts devoted to catalysts development over several decades, catalysts, especially non-precious Ni- and Co-based catalysts, have approached good activities reaching the equilibrium reaction rate but face severe coke deposition due to the unbalanced kinetics between fast C-H activation and sluggish CO₂ activation.” to “With numerous efforts devoted to catalysts development over several decades, catalysts, especially non-precious Ni- and Co-based catalysts, have approached good activities reaching the equilibrium conversion but face severe coke deposition due to the unbalanced kinetics between CH₄ activation and carbon removal”.

In the fourth paragraph of the introduction, we change “A concept of balancing kinetics of elemental steps-activation of C-H and CO₂-for overall thermodynamics optimization (BKTO) was

proposed (illustrated in Supplementary Fig. 1).” to “A concept of balancing kinetics of elemental steps-activation of C-H and carbon removal for overall thermodynamics optimization (BKTO) was proposed. (illustrated in Supplementary Fig. 1)”. We modified the following sentence in the same paragraph, from “Temperature-programmed surface reactions (TPSR) rationalized the role of Ga in achieving a more balanced kinetics between redox elementary steps by reducing CH₄ activation to match CO₂-assisted carbon removal.” to “Temperature-programmed surface reactions (TPSR) rationalized the role of Ga in achieving a more balanced kinetics between redox elementary steps by reducing CH₄ activation to match carbon removal”.

We also added the following sentences in Computational study section on Page 7 of the manuscript to further clarify the concept, “Apart from the unbalanced kinetics between carbon accumulation (via CH₄ dehydrogenation) and carbon removal, we also need to balance the carbon removal and the CO₂ dissociation process since they are both affected by the oxygen binding energy, i.e., an active O* (weak binding energy) can facilitate coke removal by O* but inhibit the CO₂ dissociation process. As a result, we need to understand and control the three processes (CH₄ activation, CO₂ dissociation, and coke removal).”

(2) The authors appear to have equated CO₂ activation with a direct reaction between CO₂ and C* on the surface. This is also conceptually inappropriate because the direct reaction between CO₂ and the carbon deposits is not supported by any mainstream studies as catalytically relevant for methane reforming on metal catalysts.

CO₂ activation typically proceeds on vacant metal sites, and the removal of C* is assisted by O* or OH* species derived from the weak or strong oxidant (CO₂, H₂O, O₂) in representative sets of reforming chemistry (DRM, steam reforming, tri-reforming, or autothermal reforming). Accordingly, SuppleFig. 1 requires to be corrected.

Reply: We agree with the reviewer that the CO₂-assisted coke removal usually involve the activation of CO₂ and that the carbon removal is usually facilitated by the surface oxygen species. While in the TPSR experiment, we measured the CO₂ consumption through an overall CO₂ reaction with C where C was purposely deposited onto the catalyst through CH₄ cracking. We, therefore, considered the possibility of reverse Boudouard reaction in our mechanism. Our computational results agree with the mainstream mechanism that CO₂ assisted coke elimination contains two elementary steps-CO₂ activation and O* reaction with C*.

Although our DFT study and mainstream mechanism consider CO₂ activation as the major route of CO₂ assisted coke elimination, we do not have solid experimental evidence supporting this route. Therefore, we considered both CO₂ activation and reverse Boudouard reaction when constructing the schematic mechanism. We used dashes instead of solid lines to indicate the uncertainty in the mechanism.

We changed the Supplementary Fig. 1.

Supplementary Fig. 1 | Schematic illustration of the design concept of coke-free dry reforming of methane catalyst through balancing kinetics for thermodynamic optimization (BKTO). The optimized catalyst has the balanced kinetics between CH_4 dehydrogenation rate (k_1) and coke removal rate (k_2). We considered both CO_2 activation followed by C^* removal with O^* route and reverse Boudouard reaction route as CO_2 -assisted coke elimination mechanism. Black ball: carbon; grey ball: hydrogen; blue ball: oxygen; and green stone: $\text{CoAl}_x\text{Ga}_{(1-x)}\text{O}_4\text{-R}$.

(3) There does not seem to be a significant difference in the catalytic stability among the CoAl spinel oxide catalysts shown in Fig. 1. For the CoAl_2O_4 catalyst, the deactivation mainly occurred at the initial 15 h, which could be due to some acid-catalyzed carbon deposition on the support, i.e., not necessarily associated with coke deposition on metal particles and the catalyst stability. The authors need to check the carbon balance before and after this initial stage.

Reply: The challenge in designing a stable DRM catalyst is to suppress coke deposition, which is also what we are pursuing in this work. Carbon balance was estimated through the downstream CH_4 , CO_2 , and CO concentrations measured by GC⁵. The carbon balance analysis (Figure R1) suggests that the coke formation on the $\text{CoAl}_2\text{O}_4\text{-R}$ happened over the entire 100-h test instead of the initial 15 h. The major coke species formed on the $\text{CoAl}_2\text{O}_4\text{-R}$ is carbon nanotubes which may not have a significant impact on its stability.

$$\text{Carbon balance (\%)} = \frac{[\text{CH}_4]^{\text{outlet}} + [\text{CO}_2]^{\text{outlet}} + [\text{CO}]^{\text{outlet}}}{[\text{CH}_4]^{\text{inlet}} + [\text{CO}_2]^{\text{inlet}}}$$

Figure R1. Carbon balance calculation during 100 h stability test of CoAl₂O₄-R.

(4) The 1000 h test is good to have and the absence of coke deposits may be a remarkable finding. From the Raman spectra and the TPO profile in Figure S4a, it can be indeed confirmed that there is no carbon deposition on the CoAlGaO₄ catalyst. However, this catalyst exhibited a similar stability as the CoAl₂O₄ catalyst, or perhaps somewhat inferior (Fig. 1). The additional CoAlMnO₄-R and CoAlFeO₄-R catalysts also showed visible deactivation, without detectable coke deposition. Thus, given the lack of a compelling relationship between the stability data and TPO/Raman characterization results, I'm left wondering what really is the decisive evidence that links the measured carbon deposits to the observed stability?

Reply: We appreciate the reviewer's acknowledgment about this potentially remarkable finding based on our 1000 h stability test. Our experimental results suggest that a significant portion of the coke formed on CoAl₂O₄ exists in the form of carbon nanotubes. While a small quantity of carbon nanotubes may not have a significant impact on overall stability, prolonged stability testing will eventually lead to catalyst deactivation and detrimental reactor blockage. To demonstrate the prevention of coke deposition, we introduced CoAlMnO₄-R and CoAlFeO₄-R as evidence, where Al was substituted with other M³⁺ metals instead of Ga.

However, it is important to note that catalyst stability is not solely determined by coke deposition. As mentioned earlier, the deactivation of a catalyst can be induced by structural changes within the catalyst itself, but this is beyond the scope of this work focused on reducing coke deposition through reaction kinetics. Coke deposition poses a substantial challenge in industrial applications, particularly in fixed-bed reactors. The accumulation of coke, regardless of its form, can obstruct or even damage the reactor, necessitating the interruption of the reaction and significantly increasing the capital cost of operation.

We revised the sentence on Page 9 of the manuscript, "It is worth noting that the deactivation of CoAlMO₄-R can be induced by other factors, such as changes in surface structure, which can be

further improved by changing Al to M^{III} ratio, modifying pre-treatment conditions, and tuning DRM reaction conditions to enhance stability while maintaining coke-free”.

(5) DFT calculations: “These increases in the barrier of the later C-H bond activation suggest that adding Ga could suppress coke formation by inhibiting deep dehydrogenation, which is consistent with observed CH₄-TPSR in Fig. 3a.” By “increases in the barrier of the later C-H bond activation”, do the authors mean that the formation of *CH₂O before the cleavage of C-H of *CH₂? Then, what is the barrier energy for the cleavage of C-H of the formed *CH₂O and the following *CHO?

Reply: We want to clarify that “the later C-H bond activation” is used for the 3rd and 4th C-H bond cleavage. We did not include the barrier of the breaking C-H bond of CH₂* and CH* via O-assisted pathway for CoGa_x surface since the direct C-H bond cleavage seems to be the favorable pathway on Co surface.

We are aware of the O-assisted C-H activation in methane (J. Am. Chem. Soc. 2017, 139, 20, 6928–6945; Appl. Catal., B 270, 118859 (2020)) and discussed it briefly in the main text. For the 1st C-H bond cleavage, our DFT calculations show that a higher activation energy is required for O*-assisted pathway than the direct C-H bond cleavage pathway (i.e., 3.0 vs 1.0 eV). For the 4th C-H bond cleavage, it was reported that the barrier was similar on Co for both pathways (Appl. Catal., B 270, 118859 (2020)). Since the 1st or the 4th C-H bond breaking is typically the rate-limiting step, the higher activation energies of these steps of O*-assisted pathway make it a less competitive pathway on the Co surface. In addition, the high coverage of reactive oxygen O* on the catalyst surface is reported to have a significant role for the O*-assisted pathway (J. Am. Chem. Soc. 2017, 139, 20, 6928–6945). However, the higher barrier of CO₂ dissociation on CoGa_x could lead to reduced availability of O* on CoGa_x as compared to the Co surface. Hence, it is expected that the O*-assisted pathway is less pronounced on CoGa_x.

Moreover, from the DFT-computed energy barriers, the introduction of Ga did not suppress but promote the activation of methane (the first C-H bond activation is typically rate-limiting), while the activation of CO₂ or reverse Boudouard reaction was suppressed (Fig. 4c). I’m afraid I cannot see any consistency here.

Reply: For the CH₄ activation process on Co surface, the rate-limiting step (RLS) is typically the 1st C-H bond cleavage as the reviewer also mentioned. However, on the CoGa_x surface, the RLS of the CH₄ activation process is likely the 4th C-H bond activation. Hence, despite promoting of the 1st C-H bond activation, introducing Ga suppresses the CH₄ activation process due to the increase of the barrier of the 3rd and 4th C-H bond cleavage (Supplementary Table 1). A similar increase of activation barriers was observed in our DFT calculations for CO₂ dissociation and the reverse Boudouard reaction. As a result, the activation barriers of these two reactions (CH₄ and CO₂ activation) become comparable. Instead, the carbon removal barrier is reduced after Ga incorporation, thus help to suppress carbon accumulation on the surface.

We revised the main text on Page 6 and 8 to clarify it as follows:

“It is generally accepted that in DRM reaction, CH₄ deep dehydrogenation and the Boudouard reaction (CO disproportionation) are the two main sources of coke formation, while CO₂ assists carbon elimination via the reverse Boudouard reaction.”

“Hence, despite promoting the 1st C-H bond activation, introducing Ga suppresses coke formation by inhibiting deep dehydrogenation due to the increase of the barrier of the 3rd and 4th C-H bond cleavage, which is consistent with observed CH₄-TPSR in Fig. 3a.”

“However, a high coverage of reactive oxygen O* on the catalyst surface is needed to have a significant role for the O*-assisted pathway. Hence, it is expected that the O*-assisted pathway is less pronounced on CoGa_x.”

(6) This work contains a few inappropriate expressions. For example, “... have approached good activities reaching the equilibrium reaction rate...”. What on earth is “equilibrium reaction rate”? At equilibrium, net reaction rate is zero and there is no way to determine the forward and reverse rate. Therefore, there is no such thing that can be called an “equilibrium reaction rate”.

Reply: Thanks reviewer for pointing out this issue. We changed the “equilibrium reaction rate” to “equilibrium conversion”.

(7) The peaks for the Ga-rich samples seem too small to be identified as meaningful information (Fig. 3).

Reply: We understand the reviewer’s concern, but the CH₄ and CO₂ consumption peaks observed in the TPSR experiment are substantial and cannot be dismissed as noise or artificial errors, with the exception of CoGa₂O₄, for which we inadvertently omitted the TPSR temperature in the comparison. To facilitate the readers' comprehension of our manuscript, we have zoomed in on the mass spectrum signals in Figure 3:

Fig. 3 | Coke resistance mechanism study. (a) CH₄-TPSR, mass signal m/z=15; (b) CO₂-TPSR, mass signal m/z=44, and (c) CH₄ and CO₂ consumption peak temperatures; peak difference between CO₂ and CH₄: CoAl₂O₄-R – 173.9 °C, CoAl_{1.5}Ga_{0.5}O₄-R – 169.0 °C, CoAlGaO₄-R – 143.0 °C, and CoAl_{0.5}Ga_{1.5}O₄-R – 69.4 °C.

(8) The results of CO₂-TPSR cannot be adequately explained without knowing the manner of CO₂ activation. From your DFT calculation results (Figure S11), the direct dissociation of CO₂ to CO and C can better explain the CO₂-TPSR data, but not the reverse Boudouard reaction.

Reply: We agree with the review that a clear CO₂ activation manner is necessary to have deep insight in the CO₂-TPSR. We want to emphasize that the CO₂-TPSR was conducted on the spent catalyst following the CH₄-TPSR. As the CH₄-TPSR leads to the coke deposition on the catalyst surface, the presence of both CO₂ and C* during the CO₂-TPSR allows parallel reactions of CO₂ dissociation and the reverse Boudouard reaction, both of which lead to CO₂ activation. In the DFT calculations, we find that the activation barriers of both reactions increase when Ga is introduced. As the CO₂ dissociation has a lower barrier over Ga and CoGa, we agree that the increased onset temperature in the CO₂-TPSR could be better explained by the increased barrier for CO₂ dissociation, but we cannot completely exclude the possibility of reverse Boudouard reaction. We, therefore, prefer to use CO₂-assisted coke elimination to describe and discuss the CO₂-TPSR study. In our manuscript, we complemented the TPSR with a computational study to elucidate the mechanism of CO₂-assisted coke elimination thoroughly and comprehensively, an aspect that the experimental study alone struggles to address adequately. We have revised the main text on Page 6 and 7 to clarify this point.

We made the following change in the main text:

In the first paragraph of “Experimental mechanism study” section, we changed “Therefore, a CH₄-temperature programmed surface reaction (TPSR) experiment followed by CO₂-TPSR was performed to test the CH₄ activation and CO₂ dissociation properties on the spinel oxides derived catalysts.” to “Therefore, a CH₄-temperature programmed surface reaction (TPSR) experiment followed by CO₂-TPSR was performed to test the CH₄ activation and CO₂ assisted coke removal properties on the spinel oxides derived catalysts.”

In the second paragraph of “Experimental mechanism study” section, we change “Similar to CH₄ activation, the CO₂ dissociation temperature rises with the addition of Ga, but to a much smaller extent, as shown in Fig. 3b. Fig. 3c shows that the peak temperature differences between CH₄ activation and CO₂-assisted coke elimination decrease from 173.9 °C on CoAl₂O₄-R to 69.4 °C on CoAl_{0.5}Ga_{1.5}O₄-R. A higher CH₄ activation temperature suggests a slower coke deposition rate, while a lower CO₂ reduction temperature indicates better coke elimination. With the smallest temperature difference between these two reactions, CoAl_{0.5}Ga_{1.5}O₄-R has a more balanced redox chemistry over the surface, leading to better coke resistance and stability enhancement.” to “Similar to CH₄ activation, the CO₂ consumption temperature rises with the addition of Ga, but to a much smaller extent, as shown in Fig. 3b. Fig. 3c shows that the peak temperature differences between CH₄ activation and CO₂-assisted coke elimination decrease from 173.9 °C on CoAl₂O₄-R to 69.4 °C on CoAl_{0.5}Ga_{1.5}O₄-R. A higher CH₄ activation temperature suggests a more difficult coke deposition. With the smallest temperature difference between these two reactions, CoAl_{0.5}Ga_{1.5}O₄-R has a more balanced redox chemistry over the surface, leading to better coke resistance and stability enhancement”.

In the first paragraph of computational study , we changed “The coke deposition process was analyzed via the dehydrogenation of CH₄ to C, the disproportionation of CO, and carbon diffusion; likewise, the coke removal process was evaluated via the reverse-Boudouard reaction (C + CO₂ ⇌ 2CO), and carbon direct oxidation reaction (C + O → CO) (Fig. 4).” to “The coke deposition process was analyzed via the dehydrogenation of CH₄ to C, the disproportionation of CO (2CO ⇌ C + CO₂), and carbon diffusion; likewise, the coke removal process was evaluated via carbon direct oxidation reaction (C + O → CO). The reverse-Boudouard reaction (C + CO₂ ⇌ 2CO) was investigated for CO₂ activation and carbon removal (Fig. 4).”

(9) This reviewer is unsure whether CoAl₂O₄ could be said to be “a previously well-studied active DRM catalyst”. The authors may want to provide some good citations that would support this claim. To prove that it is really a catalyst with good activity, the authors should calculate the reaction rate (which should be the forward reaction rate, not the net reaction rate), rather than conversion of methane. Then it would be better to put such an activity into perspective, i.e., comparing with other catalysts reported.

Reply: We changed “well-studied” to “reported” on Page 2 of the manuscript. From our experimental results, CoAl₂O₄-R is a good DRM catalyst with near equilibrium conversion (Fig. 1 and Supplementary Fig. 2). However, we believe that diverting our attention to CoAl₂O₄ detracts from the main narrative of this study. While it is certainly important to determine the "intrinsic" reaction rate or activity of the active catalyst and compare it with other catalysts in the field, our primary focus in this work is on increasing resistance to carbon deposition, which is currently a more critical issue in the DRM field. For comparison purposes, we calculated the net reaction rate of CoAl₂O₄: 0.87 mol_{CH₄} g_{cat}⁻¹ h⁻¹ and 1.07 mol_{CO₂} g_{cat}⁻¹ h⁻¹ , and these values can be compared with literature summarized in Supplementary Table 2.

- 1 Wang, Y., Hu, P., Yang, J., Zhu, Y. A. & Chen. C-H bond activation in light alkanes: a theoretical perspective. *Chemical Society reviews* (2021). <https://doi.org/10.1039/d0cs01262a>
- 2 Guo, J., Lou, H. & Zheng, X. The deposition of coke from methane on a Ni/MgAl₂O₄ catalyst. *Carbon* **45**, 1314-1321 (2007). [https://doi.org:https://doi.org/10.1016/j.carbon.2007.01.011](https://doi.org/https://doi.org/10.1016/j.carbon.2007.01.011)
- 3 Liu, J.-X., Su, H.-Y., Sun, D.-P., Zhang, B.-Y. & Li, W.-X. Crystallographic Dependence of CO Activation on Cobalt Catalysts: HCP versus FCC. *J. Am. Chem. Soc.* **135**, 16284-16287 (2013). <https://doi.org/10.1021/ja408521w>
- 4 Pei, Y.-P. *et al.* High Alcohols Synthesis via Fischer–Tropsch Reaction at Cobalt Metal/Carbide Interface. *ACS Catal.* **5**, 3620-3624 (2015). <https://doi.org/10.1021/acscatal.5b00791>
- 5 Akri, M. *et al.* Atomically dispersed nickel as coke-resistant active sites for methane dry reforming. *Nat Commun* **10**, 5181 (2019). <https://doi.org/10.1038/s41467-019-12843-w>

REVIEWER COMMENTS

Reviewer #2 (Remarks to the Author):

I can see the authors have made proper amendments to the erroneous statements such as “sluggish CO₂ activation”. I find most of their replies reasonable but have more comments to offer:

(1) What I still dislike about the paper is that they insist they have developed “a neoteric concept to design coke-free catalysts by balancing kinetics of elementary steps for the overall thermodynamics optimization (BKTO)”. To be clear, I’m not questioning the utility of optimizing the thermodynamics of key elementary steps via catalyst design, but I do not think the so-called BKTO concept captures what the authors have really done in this work. At best, they managed to discover a catalyst that was very stable over a long period of time in dry reforming of methane, but note that this was achieved at the expense of the catalytic propensity for activating C-H bonds in methane. In fact, the catalytic activity was not good enough. In view of this, I am not sure why the present authors are obsessed with the idea of putting forward a not-so-useful concept as BKTO (I doubt if it even qualifies as a concept?) to rationalize their finding?

(2) Similarly, I have a problem with the title “suppressing C-H activation promotes coke-free dry reforming of methane”, which insinuates that i) suppressing C-H bond activation is the way to go toward designing more stable catalysts and ii) suppressing C-H activation enhances the dry reforming rates (the word “promote” carries an ambiguous meaning). If that is what the authors really meant, then I decidedly disagree, because suppressing C-H activation for a reaction that is kinetically limited by C-H bond activation is certainly not an ideal approach. Apparently, no one would really love the idea of trading a substantial portion of the catalytic activity for stability, if they had a different choice without having to compromise the catalytic activity. In this regard, discovering a catalyst that would promote C* removal instead should be the more obvious way to go, no? Based on the current mechanistic understanding in the mainstream literature on dry reforming, the benefit would be maximized, ideally if the ability to scavenge C* from CH₄ dissociation is improved to such an extent that matches the C* formation rate, while not compromising or better yet, even enhancing the C-H bond activation.

(3) As I commented in the previous round of review, the 1000 h stability data was impressive but let us note that it was achieved with a diluted CH₄-CO₂ stream and relatively low conversions (CH₄: 40%). Many prior works have achieved a decent stability over a long period of time during the dry reforming of methane with higher partial pressures and conversions, without deliberately suppressing the C-H bond activation (see, for example, *Catalysis Today* 2001, 68, 217-225); some of these have found appropriate descriptors for the deactivation rate constant/stability (*Journal of Catalysis*, 2017, 356, 147). I suggest the authors to collect stability data at higher conversions and partial pressures.

(4) Figure R1 in the reply: The authors were asked to present carbon balance analysis on all the catalysts shown in Figure 1a. Here’s one now, but where are the others? My overall impression about the understanding of the relation between deactivation and carbon deposition in this paper is

not favorable; though I agree that other factors also contribute to deactivation, I do not see independent evidence that suggests the interplay of these factors and their influence on deactivation.

In total, I think the only highlight of the paper is the long-term stability of the catalyst, which was, however, achieved under conditions that may not be sufficient to speak for a decisive edge over the reported systems in the literature. Too often, the different conditions used across different studies prevent a direct and fair comparison. Moreover, I cannot appreciate the scientific significance of the BKTO concept and its relation to the major representation of the research data in this study. Therefore, I remain negative about its suitability for publication at Nature Communications.

Response to Reviewers' Comments:

We have revised the manuscript and responded to all questions raised by the reviewer. The original questions are colored **black**, our answers are in **blue**, and the revisions to the manuscript are in **red**.

(1) What I still dislike about the paper is that they insist they have developed “a neoteric concept to design coke-free catalysts by balancing kinetics of elementary steps for the overall thermodynamics optimization (BKTO)”. To be clear, I’m not questioning the utility of optimizing the thermodynamics of key elementary steps via catalyst design, but I do not think the so-called BKTO concept captures what the authors have really done in this work. At best, they managed to discover a catalyst that was very stable over a long period of time in dry reforming of methane, but note that this was achieved at the expense of the catalytic propensity for activating C-H bonds in methane. In fact, the catalytic activity was not good enough. In view of this, I am not sure why the present authors are obsessed with the idea of putting forward a not-so-useful concept as BKTO (I doubt if it even qualifies as a concept?) to rationalize their finding?

Response:

Following the reviewer’s suggestion, we removed the BKTO concept throughout the manuscript and focused more on the catalyst development and mechanistic understanding of the coke-free DRM. Through this study on coke-free DRM, we demonstrated to the scientific community the valuable potential of incorporating slower kinetics into the toolkit for catalytic process optimization. By deliberately modulating the kinetics of C-H activation, we can effectively enhance the overall performance. For a comprehensive assessment, we have included a comparative table (Supplemental Table 2) that juxtaposes our findings with the latest developments in coke-free catalysts. Of significance is the fact that none of the presently known coke-free catalysts achieve a CH₄ reaction rate $>0.5 \text{ mol } g_{\text{cat}}^{-1} \text{ h}^{-1}$, a performance threshold that our coke-free catalyst surpasses. This outcome, besides the stability over 1000 h, underscores the substantial advancements our catalyst brings to the field.

Supplementary Table 2 | Performance comparison of DRM catalysts.

Catalyst	Flow	Temp. (°C)	Rate ($\text{mol } g_{\text{cat}}^{-1} \text{ h}^{-1}$)	Coke formation	Reference
CoAl_{0.5}Ga_{1.5}O₄-R	300 L $g_{\text{cat}}^{-1} \text{ h}^{-1}$ CH₄:CO₂:He=1:1:8	700	CH₄: 0.51 CO₂: 0.70	1000 h coke-free	This work
Mo doped Ni on single crystal MgO ^b	60 L $g_{\text{cat}}^{-1} \text{ h}^{-1}$ CH ₄ :CO ₂ :He=1:1:8	800	CH ₄ : ~0.27 CO ₂ : ~0.27	850 h coke-free	(3)
Ni atomically dispersed over CeOx doped hydroxyapatite	60 L $g_{\text{cat}}^{-1} \text{ h}^{-1}$ CH ₄ :CO ₂ :He=1:1:3	750	CH ₄ : 196.4 CO ₂ : 330.1	100 h ~2% weight loss by TGA	(4)

Ni@BO _x /h-BN	60 L $g_{cat}^{-1} h^{-1}$ CH ₄ :CO ₂ :N ₂ =2:2:1	750	CH ₄ : 0.78 CO ₂ : 0.86	40 h No coke from TEM&SEM	(5)
Multielement oxide layer confined Ni	30 L $g_{cat}^{-1} h^{-1}$ CH ₄ :CO ₂ :N ₂ =1:1:3	800	CH ₄ : ~0.19 to 0.21 CO ₂ : ~0.20 to 0.22	300 h coke-free	(6)
Ni/ZrO ₂ @BN	25 L $g_{cat}^{-1} h^{-1}$ CH ₄ :CO ₂ =1:1	750	CH ₄ : ~0.39 CO ₂ : ~0.45	200 h 2.8 wt.% coke	(7)
Ni/Ce _{0.9} Eu _{0.1} O _{1.95}	60 L $g_{cat}^{-1} h^{-1}$ CH ₄ :CO ₂ :N ₂ =1:1:2	600	CH ₄ : 0.16 CO ₂ : 0.23	700 min 18.7% weight loss from TGA	(8)
Ni@HZSM-5	690 L $g_{cat}^{-1} h^{-1}$ 33.0% CO ₂ /10.6% CH 4/2.6% Ar/53.8% He	500	CO formation: 0.348 $mol_{CO} g_{Ni}^{-1} h^{-1}$	20 h ~0.43 wt.% coke	(9)

(2) Similarly, I have a problem with the title “suppressing C-H activation promotes coke-free dry reforming of methane”, which insinuates that i) suppressing C-H bond activation is the way to go toward designing more stable catalysts and ii) suppressing C-H activation enhances the dry reforming rates (the word “promote” carries an ambiguous meaning). If that is what the authors really meant, then I decidedly disagree, because suppressing C-H activation for a reaction that is kinetically limited by C-H bond activation is certainly not an ideal approach. Apparently, no one would really love the idea of trading a substantial portion of the catalytic activity for stability, if they had a different choice without having to compromise the catalytic activity. In this regard, discovering a catalyst that would promote C* removal instead should be the more obvious way to go, no? Based on the current mechanistic understanding in the mainstream literature on dry reforming, the benefit would be maximized, ideally if the ability to scavenge C* from CH₄ dissociation is improved to such an extent that matches the C* formation rate, while not compromising or better yet, even enhancing the C-H bond activation.

Response:

We agree with the reviewer and revised the title to “**Balancing elementary steps enables coke-free dry reforming of methane**” to avoid misleading part of the current title. Upon examining recent developments in the field of DRM, it becomes apparent that achieving a genuinely coke-free process poses considerable challenges. This work, alongside many others, has achieved commendable activity levels that approach equilibrium conversion rates. While researchers, including the authors, have actively explored strategies to enhance the removal of carbonaceous deposits (C*), it's crucial to recognize that executing this endeavor is notably more complex. Only a few researchers have successfully identified catalysts capable of effectively promoting C* removal, often relying on precious metals like iridium (Ir) as a catalyst base. However, this approach significantly escalates the production costs of catalysts, making them less practical for widespread industrial implementation. This current study, however, uses a relatively simple synthesis process with earth-abundant materials, and paves the way to achieve a coke-free catalyst. In addition, with the current state of DRM catalysts, the main challenge is the low

overall performance of the catalyst due to the low stability of the catalyst, rather than catalyst activity. Intending to enhance overall performance, it is more applicable on an industrial scale to redesign catalysts for prolonged stability to increase the overall performance.

(3) As I commented in the previous round of review, the 1000 h stability data was impressive but let us note that it was achieved with a diluted CH₄-CO₂ stream and relatively low conversions (CH₄: 40%). Many prior works have achieved a decent stability over a long period of time during the dry reforming of methane with higher partial pressures and conversions, without deliberately suppressing the C-H bond activation (see, for example, *Catalysis Today* 2001, 68, 217-225); some of these have found appropriate descriptors for the deactivation rate constant/stability (*Journal of Catalysis*, 2017, 356, 147). I suggest the authors to collect stability data at higher conversions and partial pressures.

Response:

We have effectively demonstrated the capabilities of the CoAl_{0.5}Ga_{1.5}O₄-R catalyst under conditions of high conversion. This is evident in Supplementary Figure 2, where CoAl_{0.5}Ga_{1.5}O₄ achieves CH₄ conversions exceeding 70%. Unlike the stability tests reported in many other works, we performed the stability test at 700 °C, which is more favorable for coke deposition compared to higher temperatures, such as 750 °C and 800 °C. As we underscored in the manuscript, our deliberate choice to conduct stability testing at low conversion, within a kinetically controlled regime, was aimed at enhancing our ability to discern deactivation trends, as accepted in the catalysis community (*Nature Catal.* 3, 471–472, 2020). Our belief is that performing stability tests at near-equilibrium conversions, as illustrated in the articles provided in the comments, can be misguided. When operating at or close to equilibrium conversion, the excess of active sites can mask any loss of activity or active site deactivation. Consequently, stability tests conducted under such conditions might yield erroneous conclusions and be of limited significance.

(4) Figure R1 in the reply: The authors were asked to present carbon balance analysis on all the catalysts shown in Figure 1a. Here's one now, but where are the others? My overall impression about the understanding of the relation between deactivation and carbon deposition in this paper is not favorable; though I agree that other factors also contribute to deactivation, I do not see independent evidence that suggests the interplay of these factors and their influence on deactivation.

Response:

We kindly remind that the primary focus of this work does not center on an in-depth exploration of all the factors leading to deactivation. While investigating the reasons behind deactivation would indeed be intriguing and valuable, it is important to note that such an endeavor lies beyond the scope of our current study centered on the balancing kinetics approach for achieving coke-free DRM. As we emphasized in the introduction part, preventing carbon deposition is important for industrial applications. We appreciate the suggestion for potential follow-up research in this direction, which could shed light on the deactivation mechanisms; however, it is important to emphasize that our immediate objective revolves around the balancing kinetics methodology and its application in the context of coke-free DRM.

In total, I think the only highlight of the paper is the long-term stability of the catalyst, which was, however, achieved under conditions that may not be sufficient to speak for a decisive edge over the reported systems in the literature. Too often, the different conditions used across different studies prevent a direct and fair comparison. Moreover, I cannot appreciate the scientific significance of the BKTO concept and its relation to the major representation of the research data in this study. Therefore, I remain negative about its suitability for publication at Nature Communications.

Response:

We appreciate all the comments and suggestions raised by the reviewer. Based on our responses above, this work proposes an approach that leads to improvement of the overall performance of a DRM catalyst, with superior CH₄ and CO₂ conversion rates when compared to the cutting-edge coke-free catalysts available in the literature and, more importantly, stable conversion without coke formation. We agree that this work hasn't solved all the challenges toward industrial application, but the concept, together with the facile method to synthesize the materials, lays a foundation for advancing this important C1 chemistry.